# Context Matters: Preliminary Evidence That the Association between Positive Affect and Adiposity in Infancy Varies in Social vs. Non-Social Situations

**DOI:** 10.3390/nu14122391

**Published:** 2022-06-09

**Authors:** Alexis C. Wood, Shabnam R. Momin, MacKenzie K. Senn, David J. Bridgett

**Affiliations:** 1USDA/ARS Children’s Nutrition Research Center, 1100 Bates Avenue, Houston, TX 77030, USA; shabnam.momin@bcm.edu (S.R.M.); mackenzie.senn@bcm.edu (M.K.S.); 2Department of Psychology, Northern Illinois University, DeKalb, IL 60115, USA; dbridgett1@niu.edu

**Keywords:** infancy, adiposity, positive affect, temperament, longitudinal, situational effects, BMI, weight-for-length, context, development

## Abstract

Previous studies have suggested that infants high in negative affect have higher levels of adiposity, arising in part via changes in nutrition (e.g., “feeding to soothe”). Few studies have examined whether positive affect shows similar or inverse associations with adiposity. The current study examined cross-sectional and longitudinal relationships between adiposity and observations of positive affect in both a social and a non-social context, using data from infants at four (*n* = 125) and 12 (*n* = 80) months of age. Our analyses did not find any cross-sectional associations between positive affect and adiposity (all *p* > 0.05). However, in the longitudinal analyses, positive affect in a non-social context, when observed at four months of age, was positively associated with weight-for-length at 12 months of age (zWFL; ß = 1.49, SE = 0.67, *p* = 0.03), while positive affect observed at four months of age in a social context was inversely associated with body fat percentage at 12 months of age (ß = −11.41, SE = 5.44, *p* = 0.04). These findings provide preliminary evidence that the p positive affect is related to adiposity in infancy and suggest that the direction of association (i.e., direct or inverse) may be specific to the context in which positive affect is measured. Future research should examine the role of nutritional status in any relationships between adiposity and emotion at this early stage.

## 1. Introduction

Approximately 16% of infants in the United States (US) show a weight for length [WFL] greater than the 95th percentile for their age and gender at 12 months of age [1]. High WFL at the end of the first year of life is associated with increased risk for obesity in early childhood [2], placing these children at an elevated risk both for obesity across the lifespan [2] and for weight-related medical, metabolic, and psychological comorbidities in childhood including Type 1 diabetes, asthma, dental caries, high blood pressure, poorer glucose control, and both internalizing and externalizing disorders [3,4]. Because the accrual of adiposity occurs over time, to prevent the onset of excess adiposity, we need to identify factors that influence rapid weight gain at the earliest ages—ideally within the first six months of life -- yet, we know little about which factors influence adiposity trajectories during this time. High maternal body mass index (BMI) and poor metabolic health before [5,6,7] and during [5,7,8] pregnancy, and subsequent infant birth weight [7,9], existing risk factors for accelerated weight gain in infancy, together only explain a small percentage of individual differences in infant weight gain [9]. To date, there remains a lack an understanding of who is at risk for accelerated early weight gain, particularly during the earliest months of life. 

Obesity risk reflects an interplay between an individual’s genetic predispositions and their environment across the lifespan [10,11,12], including during early infancy [13,14,15]. As in older ages, the role of stressful environments such as socioeconomic challenges [16] and a lack of warm/responsive parenting [17,18] on weight gain during the first year of life are fairly well established, but infant characteristics that contribute to adiposity are less well identified. Historically, aspects of infant temperament have been considered a likely source of individual differences in early adiposity. Defining temperament is a contentious issue, and different conceptual frameworks imply different definitions of temperament during infancy. However, there is general acceptance that temperament during the first year of life encapsulates an interplay between two constructs: (1) reactivity, which refers to the arousability of sensory, motor, and affective systems [19], and (2) self-regulation, which refers to the use of internal processes to modulate reactivity and behavior.

Self-regulation has received considerable attention for its relationship to obesity than reactivity, with a substantial body of literature demonstrating inverse associations between self-regulation and adiposity across toddlerhood [20,21,22], early and middle childhood [23,24,25,26,27,28,29,30,31,32,33,34,35,36,37,38,39,40,41,42,43], adolescence [33,41,44,45,46,47,48,49,50,51,52,53,54,55,56,57,58,59,60,61,62,63], and adulthood [36,63,64,65,66,67,68,69,70,71,72,73,74,75,76,77,78,79,80,81,82,83,84,85,86,87,88,89,90,91,92,93,94,95,96,97,98,99,100,101], a pathway that likely involves contributions from nutrition differences associated with infant temperament (see [102] for a review). However, during early infancy, reactivity is modulated externally, principally by the actions of caregivers (e.g., by rocking and holding the child in order to down-regulate negative reactivity) [103,104,105,106]). Infants’ abilities to self-regulate their affective states only reach the level of maturity needed for emotional and behavioral modulation at around nine to 12 months of age. Thus, self-regulation is an unlikely source of major obesity risk during early infancy. Unlike those processes that regulate emotion, the early emotional processes themselves (i.e., positive and negative affect) are comparatively more developed in early infancy [107,108,109], yet their potential role in obesity has received relatively less attention.

Of the two emotional tones in infancy, negative affect has received more attention than positive affect in regards to its association with adiposity during the first year of life [110,111,112,113,114,115] and beyond [116,117,118], in part because of its theorized effects on caregiver feeding behaviors, i.e., “feeding to soothe” [102,106]. Yet, positive affect also influences infant development in ways that may be critical to understanding obesity risk. For example, positive affect is a developmental antecedent of self-regulation [107,119,120,121,122]—a known obesity risk factor—highlighting its role in shaping intrapersonal traits that influence adiposity. Displays of positive affect in infancy also elicit the warm and responsive parenting behaviors thought to reduce obesity risk in childhood [123], demonstrating that positive affect may associate with adiposity by shaping interpresonal risk factors. However, the association between positive affect and development may depend on context, and specifically whether the positive affect is elicited in the presence vs. the absence of the caregiver, which we label social (i.e., in the context of caregiver interaction) and non-social (i.e., displayed in the absence of the primary caregiver), respectively. These differential associations of social and non-social positive affect with development may arise because infants’ emotional development is first learned in the context of the parent-infant relationship [124]. Positive affect displayed in the presence of a caregiver is more likely to elicit the warm/responsive parenting behaviors that foster the development of good self-regulation skills in later childhood [107,119,120] and further, may also protect the child against the development of obesity [123]. Conversely, positive affect elicited by tasks that do not involve mother-child interactions (non-social positive affect) have been inversely associated with later self-regulation [119,121,122], suggesting that early social positive affect supports the development of self-regulation skills, while non-social positive affect may inhibit regulatory skill development. 

Taken together, positive affect, like negative affect, is a critical driver of both internal characteristics and external factors that contribute to obesity risk in a complex relationship that is likely to be context-specific. Yet, unlike negative affect, few studies have examined the association of infant positive affect with adiposity, and it is hard to draw conclusions from those studies that have done so. Two studies reported no association between adiposity measures and smiling and laughter when assessed at six weeks [115], eight weeks [112] and/or eight months of age [115]. However, a third study did find a positive relationship between positive affect and weight gain in the first year of life using the broader temperament domain of “surgency/extraversion”, which reflects aggregate scores across high-intensity pleasure, smiling and laughter and approach (measures via excitement and positive anticipation) [125]. Differences in findings, as well as in the definition of positive affect, across the different studies, precludes firm conclusions about whether there is a relationship of positive affect to adiposity, a problem potentially exacerbated by the use of maternal reports to assess infant temperament in all three studies [126,127]. While there are advantages of maternal reports (e.g., reporting on behavior across a wider range of situations), maternal reports may contain inherent biases and can reflect maternal characteristics such as her comprehension, knowledge, and mental state (e.g., the presence of anxiety or depressive symptoms) [128,129], as well as cultural norms and expectations [130,131,132]. In addition, maternal reports may only reflect infant behavior in repose to parental behavior [128,129], and thus, may not be suitable for distinguishing between positive affect in social vs. non-social contexts.

To address our lack of understanding regarding which factors that influence infant weight gain during the first year of life, we aimed to examine the relationship between positive affect and adiposity within and across early and late infancy. Further, we aimed to use objective assessments of infant positive affect, made by trained observers in standardized situations that measured positive affect in the presence of the caregiver (positive affect in a social context) and in the absence of a caregiver (positive affect in a non-social context). Based on the limited available evidence to date, we tentatively hypothesized that positive affect in a social context would be inversely associated with adiposity, while positive affect in a non-social context would be positively associated with weight status. 

## 2. Materials and Methods

### 2.1. Participants

The sample (*n* = 126) represents communal variables from two cohorts of infants who participated in study visits at our research center in Houston, Texas, conducted by the same research team during overlapping time spans: the Baylor Infant Twins Study (BITS; *n* = 66; 31 complete twin pairs) and the Baylor Infant Orometer (BIO) study (*n* = 60), the recruitment methods for which have been described in detail elsewhere [133]. All infants were recruited via either (1) advertisements posted in pre/post-natal clinics at Texas Children’s Hospital (TCH), Texas Children’s pediatrics (TCP), and online from the social media sites of local parent groups; or (2) by approaching women in their postpartum room at TCH. Inclusion criteria included infants who were less than 4 months of age (corrected for gestational age), with a birth weight of at least 1800 g (BITS) or 2267 g (BIO), and who were free from major congenital anomalies. Exclusion criteria included infants from higher-order multiples and/or with parents with inadequate English to understand the study protocol and provide informed consent/assent.

### 2.2. Procedure

Infants and their main caregiver(s) participated in a baseline visit to our research center in Houston, Texas, conducted at four months of age (corrected for gestational age; *n* = 125). Of those that participated in the baseline visit, 111 infants were invited to participate in a follow-up visit conducted at 12-months of age (13 infants were too old by the time the follow-up protocol was initiated, and the family of one infant did not give consent to be contacted for follow-up visits after baseline). The final sample for the 12-month follow-up visit was *n* = 80, representing an attrition rate of ~28%. During both study visits, infants participated in a play-based temperament observation, and the main caregiver completed questionnaire measures in a separate room when not participating in the observation. After the temperament observations, the infants completed the assessment of body composition. 

Written assent was obtained from all parents before study procedures were conducted, and this study was reviewed and approved by the Institutional Review Board (IRB) at Baylor College of Medicine (H-36097).

### 2.3. Measures

*Positive affect. The* dimension of positive affect, called “joy/pleasure” by the developers, was assessed via the *Laboratory Temperament* Assessment Battery (*Lab**-*TAB*)* observation [134]. The goal of Lab-TAB is to provide a standardized, laboratory-based assessment of temperament across early childhood, from infancy through preschool. Lab-TAB involves several short, play-based procedures, known as “episodes”, with each designed to measure a single dimension of temperament. There are four episodes per temperament dimension, and it is recommended to include at least two episodes for each temperament dimension of interest [134]. To make coding behaviors easier, episodes are divided into shorter intervals called epochs, within which each behavior is observed.

*Non-social positive affect* was assessed via the “puppets” episode. During the puppets episode, the experimenter sat behind a table that was placed in front of the high chair. The experimenter wore two hand puppets, which started under a table, and the episode began when the puppets were brought above the table in view of the child. The experimenter then read a fast-paced, standardized dialogue that was printed on a sheet of paper (kept under the table) for easy reference, using a lively tone with different voices for each puppet. The episode contained five epochs, each lasting approximately 15 s, during which the puppets either talked to each other (first epoch), talked to each other, and then physically tickled the child on their stomach (epochs two to four), or were placed on the table, motionless and without sound, in front of the infant (epoch five, the final epoch). For each epoch, the following behaviors were coded: (1) maximum observed intensity of smiling (0 = no smiling at all; 1 = small smile, with lips slightly upturned and no involvement of cheeks or eyes; 2 = medium smile, with lips upturned, perhaps mouth open, slight bulging of cheeks, and perhaps some crinkling about the eyes; 3 = large smile, with lips stretched broadly and upturned, perhaps mouth open, definite bulging of cheeks and noticeable crinkling of eyes); (2) the presence of laughter (0 = absent; 1 = present); (3) the presence of positive vocalizations (0 = no positive vocalizations observed; 1 = presence of positively toned babbling, squealing, and similar behaviors); and (4) the extent of engagement with the toy (0 = the child appears indifferent to the puppets; 1 = the child exhibits a mostly neutral reaction to the puppets, looking at them with only mild interest; 3 = the child appears fully engaged with the puppets and seems to like the puppets, seemly engrossed in them). 

*Social positive affect* was assessed via the “peek-a-boo” episode. During peek-a-boo, a large black board was placed approximately four feet away from the child. The board had several “doors” which are closed at the start of the episode. The mother was behind the board, and the experimenter read a standardized script, again in a lively manner. There were seven epochs, each lasting about 15 s, during which the experimenter either (1) asked “where is mommy?”, knocked on a door on the board, and opened the door to reveal the mom, whereupon the mom said “peek-a-boo!” (epochs one, two, three, and six); (2) asked “where is mommy?”, knocked on a door on the board, and opened the door to reveal an empty space whereupon the experimenter said “No, she is not there! Where is mommy?” (epochs four and five); or (3) sat quietly looking at the board with mom still out of sight behind the board (epoch seven, the final epoch). For each epoch, the following behaviors were coded: (1) the intensity of smiling; (2) the presence of laughter; (3) the presence of positive vocalizations (all coded as for puppets); and (4) the presence of positive motor activity (0 = absence; 1 = banging of hands on the table, clapping, waving of arms in excitement, and/or reaching toward the board observed).

*Scoring procedures.* For each episode, behaviors were not coded if the infant was in distress. For each behavior within an episode, a mean across all epochs was taken. These scores were then standardized and re-scaled to a range of 0–1 before a mean was taken of all the behaviors within an episode. Data were excluded from one infant in distress, who conducted the positive affect observation sitting in the lap of one of their caregivers, leaving a final sample of *n* = 125 for exploring associations between positive affect and adiposity.

*Psychometric properties.* For each episode, a subset of at least 23 children were independently coded by an additional coder to assess inter-rater reliability (IRR). IRR was generally high, ranging from an intraclass correlation (icc) of 0.76 to 0.90. (Appendix A). 

*Body composition*. Recumbent length was measured using an infant stadiometer with a fixed headpiece and horizontal backboard, and an adjustable foot piece; weight was assessed using a Sartorius weight scale. Duplicate measurements were taken for both length and weight, and a mean of the two used in analyses. Adiposity was assessed via a Dual-Energy X-ray Absorptiometry (DXA). To complete the DXA measures, the infant was swaddled and placed face-up on the DXA plate. Raw scan data was converted to an image, with a quantitative measurement of the bone and body tissues taken, resulting in the estimates of total fat and lean mass. BMI was calculated as weight divided by length squared, and both BMI and WFL were converted to age- and gender-adjusted z-scores (zBMI and zWFL respectively) according to WHO reference standards [135].

*Demographics.* The main caregiver (in all cases, the mother), recorded the child’s age, gender, and gestational age via a questionnaire.

### 2.4. Analyses

All analyses were conducted in R v4.0.2 [136]. 

*Descriptive Statistics.* Demographic, temperament, and adiposity information were calculated as total number (*n*) and percentage (%) for categorical variables, and as means +/− standard deviation (sd) for continuous variables. Differences between the same measure at different ages were assessed via a Wilcoxon signed-rank test for paired data.

*Data preparation.* Scores for each Lab-TAB episode, as well as for each of zWFL, body fat percentage, and zBMI, were examined for distributions suggesting a departure from normality using the calculated skew and sample kurtosis, and a visual inspection of histograms. Variables that deviated from normality, defined as having a skew or kurtosis less than −1, or greater than +1, were transformed before being included in the analysis using a rank normal transformation with Blom constant.

*Preliminary analyses.* The association of social and non-social positive affect at each age was assessed via Pearson correlations between scores on the puppets and peek-a-boo episodes at each visit. This initial step was taken to inform the analytic approach for primary analyses. As expected, correlations between social and non-social positive affect were modest at all ages, at 4 months: r = 0.41 (95% confidence intervals; Cis: 0.25–0.55), while at 12 months, r = 0.12 (CIs: −0.11–0.33) supporting the notion that puppets and peek-a-boo measure related, but separable, aspects of positive affect. The results of this analysis informed our decision to use the social and non-social positive affect scores at each time point as separate variables within the same regression analyses.

*Associations of temperament with adiposity.* The main effects of temperament on adiposity were examined using multi-level linear regression models, with scores for each of the three adiposity measures at each time point as the dependent variables in separate models and scores for social and non-social positive affect as independent variables within the same model. All models controlled for gender as a fixed effect and family as a random effect, such that robust standard errors (SEs) are presented, which are adjusted for the clustering of data within the family. 

## 3. Results

*Sample Descriptives*. Our sample was diverse and was about equally split between male (48%) and female (52%) infants, of whom 61.5% had a parent who self-reported their race as white, 13.1% as black, and 7.4% as Asian, with 25.8% reporting having Hispanic/Latino ancestry (Table 1). Children generally expressed more joy/pleasure as they got older, with significantly higher scores at 12 months than at four months of age for non-social positive affect in the puppets episode (t = −2.90, df = 77, *p* = 0.04; Table 1), and for social positive affect in the peek-a-boo episode (t = −5.50, df = 74, *p* ≤ 0.001; Table 1). Adiposity variables were generally more stable, with only body percentage fat body percentage showing the expected decrease from 4 to 12 months of age, (t = 43, df = 39; *p* < 0.001; Table 1).

*Associations between positive affect and adiposity* In cross-sectional multivariable linear regressions which controlled for the child’s gender, as well as the clustering of data within the family, observed positive affect was not significantly associated with adiposity at either 4- or 12-months of age (all *p* > 0.05; Table 2). In similar longitudinal analyses, positive affect in a non-social context (i.e., during puppets), when observed at four months of age, was positively associated with WFL z score at 12 months (*ß* = 1.49, SE = 0.67, *p* = 0.03; Table 2); with longitudinal associations between positive affect in a non-social context and other adiposity indices showing the same direction of association, but not reaching significance. Positive affect in a social context (i.e., during peek-a-boo), when observed at four months of age, was inversely associated with body fat percentage at 12 months of age (*ß* = −11.41, SE = 5.44, *p* = 0.04; Table 2). Again, the associations of positive affect in a social context observed at 4 months of age with other adiposity measures at 12 months of age were also inverse, but did not reach significance.

## 4. Discussion

Relatively little is known about the factors which influence individual differences in adiposity during the first year of life. Infant temperament is emerging as a potential correlate of both child nutritional status [102] and infant weight trajectory. While several studies have linked high negative affect during infancy to higher adiposity [110,111,112,113,114,115,116,117,118], fewer studies have examined whether positive affect is also associated with adiposity [112,115]. Furthermore, those that have probed positive affect for its associations with early adiposity have been limited by their reliance on maternal reports of infant temperament, and a lack of distinction between the potential for context in which positive affect is displayed to influence associations [112,115,125]. The current study is the first study, of which we are aware, to examine associations between infant positive affect and adiposity using objective observations of infant positive affect, measured at two time points during the first year of life, during which positive affect is observed in two different contexts: a social context (i.e., when the infant is interacting with their caregiver) and a non-social context (i.e., in the absence of their caregiver). We did not find any cross-sectional associations between positive affect and adiposity in either context. However, we found preliminary evidence that positive affect, when observed at four months of age, was positively associated with later adiposity when the caregiver was absent and inversely associated with later adiposity when the caregiver was present.

Positive affect in response to a puppet game, which we characterize as non-social because this Lab-TAB episode is not reliant on the caregiver–infant dyad interacting, when measured at four months of age was not significantly associated with concurrent adiposity. However, this non-social positive affect, when measured at four months of age, was related to a higher standardized WFL at 12 months of age. Two previous studies have not found an association between positive affect and adiposity in infancy [112,115]. Differences between these prior findings and those of the current study may be due, in part, to the use of parent-reported temperament in prior work; observed and parent-reported positive affect are often not correlated [137] and can reflect parental characteristics such as their preference for engaging in positive play with their children [138] and may have reflected caregiver elicited, social positive affect, and hence, differed from non-social positive affect observed in the absence of the caregiver in the current study. Only one study, of which we are aware, has reported a significant association between positive affect and adiposity, an association in the same direction as the findings from the current study, i.e., a positive association between higher positive affect and higher adiposity [125]. While the current study did not probe for the mechanisms underlying this association, previous studies have suggested that non-social positive affect, elicited by tasks that do not involve mother–child interactions, are associated with lower later self-regulation [120], including when using a similar puppets game as in the current study [121,122]. Given that lower self-regulation is associated with higher adiposity by toddlerhood and possibly reduced dietary quality [20,21,22], we had expected higher positive affect elicited by puppets to be associated with higher adiposity, and present preliminary evidence in support of our hypothesis. 

We also expected that social positive affect, i.e., as measured by a peek-a-boo game involving caregiver–infant interactions at four months of age, would be inversely associated with adiposity, given the role of early positive affect in response to caregivers in supporting the development of self-regulation. We observed a similar pattern of results as with non-social positive affect, but in the opposite direction. That is, none of the cross-sectional associations were significant, although their directionality was uniformly inverse at four months of age. However, longitudinal associations between social positive affect at four months of age and later adiposity, all of which were also in the same inverse direction, did reach significance for the percentage of body fat at 12 months of age. While this finding supported our initial hypothesis, until replicated, it should be considered preliminary in the context of previous research that has shown two null associations [112,115] and one positive [125] between adiposity and caregiver rated positive effect in infancy. 

Our study had several strengths, including the use of direct observations of infant positive affect in two different contexts at two separate ages. However, the use of a more objective assessment necessitated a more modest sample size than employed in previous studies using parent-report measures of positive affect. With the potential effects this could have on minimizing statistical power, and when considering the exploratory nature of our work due to limited existing studies in this area, we made the a priori decision not to correct for multiple testing, and hence, our results should be considered preliminary until replicated in subsequent studies. Further, although we aimed to assess positive affect in highly standardized situations, future studies should examine whether our laboratory-based approach yields different results from observations in more familiar contexts, as suggested by results in older children [139]. Finally, it was beyond the scope of the current investigation to probe for mechanisms underlying the links between positive affect and adiposity. Future research should take a comprehensive approach to examine potential mechanisms, which would likely necessitate exploring the possibility of bi-directional associations between positive affect, self-regulation, and nutritional status via food intake [102].

To date, very few characteristics during the first six months of life have been identified that shape early obesity risk. Here, we present preliminary evidence that levels of positive affect expressed during a play-based situation in early infancy may indicate increased risk for higher adiposity by late infancy, but our data suggest that the directionality of this finding may be specific to the context in which games, here to the presence vs. absence of the caregiver. Given that nutrition is linked to adiposity, an important future direction would be to explore whether positive affect is also an early determinant of food intake. 

## Figures and Tables

**Table 1 nutrients-14-02391-t001:** Demographic, temperament, and adiposity information by age.

	4 Months	12 Months
**Demographic Information**		
Gender, male		
Male	61 (48%)	
Female	65 (52%)	
Race		
Asian	9 (7.4%)	
Black	16 (13.1%)	
White	75 (61.2%)	
Other or mixed race	22 (18.0%)	
Ethnicity		
Hispanic/Latino	32 (25.8%)	
Not Hispanic/Latino	92 (72.4%)	
**Positive Affect**		
Non-social *	0.55 (0.21)	0.61 (0.23)
Social ^c^	0.34 (0.22)	0.44 (0.22)
**Adiposity**		
Weight-for-length, z score	0.85 (1.10)	1.19 (0.93)
Fat mass, % *	33.88 (6.09)	29.5 (6.30)
Standardized BMI, kg/m^2^	0.06 (1.17)	0.65 (1.01)

* 4 months vs. 12 months *p* < 0.05.

**Table 2 nutrients-14-02391-t002:** Parameter estimates from multi-level linear regressions examining associations between social and non-social positive affect (independent variable) with standardized weight-for-length (zWFL), body fat percentage, and standardized body mass index (zBMI) (dependent variables) across infancy.

	WFL, z Score				Body Fat, %				BMI, z Score			
	4 Months		12 Months		4 Months		12 Months		4 Months		12 Months	
	β (SE)	*p*	β (SE)	*p*	β (SE)	*p*	β (SE)	*p*	β (SE)	*p*	β (SE)	*p*
**Positive affect at 4 months**												
Non-social	−0.32 (0.49)	0.51	**1.49** **(0.67)**	**0.03**	−0.47 (2.98)	0.88	6.76 (4.79)	0.16	−0.37 (0.49)	0.44	1.05 (0.64)	0.11
Social	−0.16 (0.63)	0.80	−1.25 (0.70)	0.08	−1.02 (3.27)	0.76	**−11.41** **(5.44)**	**0.04**	−0.13 (0.59)	0.83	−0.37 (0.72)	0.61
**Model 2: Positive affect at 12 months**												
Non-social	-	-	0.25 (0.54)	0.65	-	-	4.43 (4.08)	0.28	-	-	−0.05 (0.52)	0.92
Social	-	-	0.20 (0.90)	0.82	-	-	0.40 (3.79)	0.92	-	-	0.61 (0.71)	0.39

Note: All models control for gender as a fixed effect and family as a random effect, such that robust standard errors (SEs) are presented, which are adjusted for the clustering of data within the family. Significant results (*p* < 0.05) in bold.

## Data Availability

Data are available from the first author upon request.

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
