# Peer review of "Context Matters: Preliminary Evidence That the Association between Positive Affect and Adiposity in Infancy Varies in Social vs. Non-Social Situations"

_nutrients, 2022, doi:10.3390/nu14122391_

Round 1

Reviewer 1 Report

The article is clearly written and evaluates the possible factors influencing weight gain in the child during the first year of life by evaluating the two types of positive affect and the body composition, assuming that the non-social positive affect may be directly proportional to adiposity, showing remarkably interesting preliminary results.

To further underline the importance of this study, I suggest to include further examples of comorbidities in the part of the introduction in which the comorbidities related to obesity that can be encountered in the course of life are discussed (https://doi.org/ 10.1007/s11547-020-01146-6; 0.3389/fpsyt.2022.831358).

Author Response

Below is a point-by-point summary of the reviewers’ helpful reports. Reviewer comments are in bold. Reponses are in normal typeface. Sections from the manuscript are indented and in italics, with changes in the revised manuscript underlined for clarity.

Reviewer 1

The article is clearly written and evaluates the possible factors influencing weight gain in the child during the first year of life by evaluating the two types of positive affect and the body composition, assuming that the non-social positive affect may be directly proportional to adiposity, showing remarkably interesting preliminary results.

To further underline the importance of this study, I suggest to include further examples of comorbidities in the part of the introduction in which the comorbidities related to obesity that can be encountered in the course of life are discussed (https://doi.org/ 10.1007/s11547-020-01146-6; 0.3389/fpsyt.2022.831358).

Response: Thank you for this suggestion. We have amended the Introduction, which now states:

High WFL at the end of the first year of life is associated with increased risk for obesity in early childhood [2], placing these children at an elevated risk both for obesity across the lifespan [2], and for weight-related medical, metabolic and psychological comorbidities in childhood including Type 1 diabetes, asthma, dental caries, high blood pressure, poorer glucose control, and both internalizing and externalizing disorders [3, 4].

Reviewer 2 Report

The authors aimed to examine the relationship between positive affect and adiposity within and across early and late infancy. The methods were appropriated to answer the research question. The introduction section is very long and could be summarized, but it is not a limitation for accepting the article for publication. Although there are other factors that influence childhood adiposity, the authors make clear the intent of the study and that other factors such as food intake should be considered in future studies.

Author Response

Reviewer 2

The authors aimed to examine the relationship between positive affect and adiposity within and across early and late infancy. The methods were appropriated to answer the research question. The introduction section is very long and could be summarized, but it is not a limitation for accepting the article for publication. Although there are other factors that influence childhood adiposity, the authors make clear the intent of the study and that other factors such as food intake should be considered in future studies.

Response: Thank you for your kind comments. We have revised the Introduction, with the goal of conveying the same information in a more succinct manner.